# Health Impact and Risk Factors Affecting South and Southeast Asian Women Following Natural Disasters: A Systematic Review

**DOI:** 10.3390/ijerph182111068

**Published:** 2021-10-21

**Authors:** Syadani Riyad Fatema, Leah East, Md Shahidul Islam, Kim Usher

**Affiliations:** 1School of Health, Faculty of Medicine and Health, University of New England, Armidale, NSW 2351, Australia; least@une.edu.au (L.E.); mislam27@une.edu.au (M.S.I.); kusher@une.edu.au (K.U.); 2Department of Sociology, Noakhali Science and Technology University, Noakhali 3814, Bangladesh

**Keywords:** women’s health, physical health, mental health, risk factors, natural disasters, systematic review

## Abstract

(1) Background: Following natural disasters, women have a higher prevalence of adverse physical and mental health outcomes. Given that the South and Southeast Asia regions are highly disaster prone, a review was undertaken to identify the potential health impact and key risk factors affecting women after disasters in the countries located in South and Southeast Asia regions. (2) Methods: A systematic literature search of four databases yielded 16 studies meeting the inclusion criteria. The review was conducted according to the Preferred Reporting Items for Systematic Review and Meta-Analysis (PRISMA) guidance, between July 2008 and March 2021. (3) Results: The majority of studies reported women’s negative/poor mental health, identifying a significant association of socio-demographics, during disaster exposure, post-disaster, and pre-existing risk factors. The six most-cited influences on women’s mental health found in the reviewed literature were being female, adult age group, having no formal education, poverty or low economic status, poor physical health/physical injuries, and death of family members. Women’s health during the post-disaster period was generally reported as poor among all the countries of the South and Southeast Asia regions. (4) Conclusions: Appropriate social support and the availability of free healthcare access for women are warranted in disaster-affected areas. This review offers a valuable contribution to the knowledge of women’s health complications/challenges and associated risk factors related to disasters, essential for the development of strategies to help reduce this burden in the future. Further research is required on natural disasters to identify ways to reduce women’s health impacts after natural disasters, especially in the context of low-income and lower-middle-income countries.

## 1. Introduction

Natural disasters are adverse environmental events not directly attributable to human acts that create fear of injury, loss of property, and dislocation of residence. They include volcanic eruptions, earthquakes, floods, cyclones, droughts, and wildfires [1,2]. The Centre for Research on the Epidemiology of Disasters (2020) reported that over the past two decades, between 2000 and 2019, 7348 natural disasters were recorded, with the highest number (3068) occurring in Asia followed by America and Africa. In Asia, other than China, the most disaster-affected countries were in South and Southeast Asia, including India, the Philippines, Indonesia, and Bangladesh [3]. Additionally, in Pakistan in 2005 and Nepal in 2015, earthquakes killed 73,300 and 8969 people, respectively, causing massive injury and infrastructural damage, resulting in these being considered the world’s two most dangerous earthquakes to date [3]. Other catastrophic disasters have occurred in South and Southeast Asia in recent times, namely the 2004 Indian ocean tsunami and the 2008 cyclone that affected Myanmar. Although all individuals are at risk of detrimental impacts of a disaster, women are more vulnerable to physical and psychological health impacts compared to men [4].

### Background

The number of people affected by natural disasters has significantly increased, with women among the most vulnerable across a range of social and cultural contexts [4]. Vulnerability to natural disasters is rooted in structural challenges such as poverty, inequalities, rapid population growth, and the downstream risk of disasters such as displacement, lack of disaster management strategies, and resilience [4,5]. Recorded disaster fatalities indicate death rates of females are often much higher than that of males. For example, 90% of the 140,000 deaths and 80% of 10,000 deaths during the cyclones of 1991 and 2007 in Bangladesh were females [6]. In addition to death, the gendered impact of disasters includes higher rates of injuries, disability, chronic disease, mental illness, poverty, dependency, family conflict, and unemployment for females [7,8,9]. In addition to the vulnerabilities that women face as a consequence of natural disasters, they are also at risk of abuse and exploitation [5]. Because of these, women’s health is the prime concern usually violated as a result of disasters, particularly in low- and lower-middle-income countries [10,11,12].

Health is defined as a state of complete physical, mental, and social wellbeing and not merely the absence of disease or infirmity [13]. According to the WHO (2007; p. 1), mental health is “a state of well-being in which the individual realizes his or her own abilities, can cope with the normal stresses of life, can work productively and fruitfully, and is able to contribute to his or her community” [14]. Mental illnesses are characterised by abnormal thoughts, emotion, behaviour, and relationships, with examples including depression, anxiety, stress, schizophrenia, bipolar disorder, and post-traumatic stress disorder (PTSD) [15]. Among these disorders, anxiety and depression are the most common, which are a reaction to physiological illness and the stressors of life [16]. In the post-disaster context, health is impacted both directly (i.e., physical injury, illness) and indirectly (i.e., short term or long-term mental illness). Literature has identified that post-disaster, women experience adverse mental health consequences. For example, women have also been found to be significantly vulnerable to depression and PTSD after earthquakes in Italy, Haiti, and Wenchuan [17,18,19]. In addition, a significant association was found between bushfire exposure and mental health illnesses among Australian women [20]. Although not all women are vulnerable to mental health consequences, women have been found to be more vulnerable to developing mental illness following a natural disaster compared to men [17,19].

A variety of factors including physiological, socio-demographic, cultural, gender inequity, and disaster exposure play a vital role in affecting women’s health after natural disasters [7,8,9]. The impact on women in Asia is exacerbated by the gender inequities and expectations experienced by women in these countries. For example, they are often the ones left at home [9,10] to take care of children, other family members, and household assets. Women may also find it harder to reach safe places (such as disaster shelter homes) during disasters because of their responsibility to evacuate children and older family members [7]. Furthermore, rural women impacted by natural disasters often miss out on pre-disaster warnings, and do not have access to gender-focused awareness programmes designed for women at home [7,8]. On average, four women die for every one male death in South Asia because of natural disasters, but the gender discrepancy is much less in countries of different cultures where women enjoy more freedom [21].

Although women in general have been found vulnerable in previous disaster studies, they remain significantly more impacted in disaster-affected areas in low-income and lower-middle-income countries [9,10,11]. The reason for this outcome may be related to their existing vulnerabilities that are often ignored or not seen as a priority. Given the increasing rate of disasters in many countries, understanding the risk factors affecting women’s health has become increasingly more important to improve the effectiveness of disaster management efforts. The aim of the review was to identify and critique the evidence of the impact of natural disasters on the physical and mental health of women, and the underlying causes/risk factors affecting their health in the countries located in South and Southeast Asia regions. Through identification of the impact and risk factors, our study can assist and improve preparedness for disasters and the implementation of appropriate plans to potentially reduce the impact of disasters on women’s health and wellbeing.

## 2. Materials and Methods

### 2.1. Protocol and Registration

The results of this study have been reported according to the Preferred Reporting Items for Systematic Review and Meta-Analysis (PRISMA) guidance 2020 (Appendix A) [22]. The review protocol is registered with PROSPERO (registration no. CRD42019123809).

### 2.2. Inclusion Criteria

The inclusion criteria for the review comprised empirical peer-reviewed studies published between July 2008 and March 2021 to return contemporary evidence, reporting on South and Southeast Asian women’s risk factors and physical and mental health outcomes following natural disasters, including women aged 18 years and above, and published in English. 

Studies were excluded if they did not address the impact of natural disasters on women’s health and studies that did not distinguish between health impacts of natural disasters and existing conditions. 

### 2.3. Search Strategy

After auditing PROSPERO to avoid unnecessary repetition, a computerised search of four databases was conducted: ProQuest, ProQuest Health and Medicine, PubMed, and EBSCO. Table 1 illustrates search terms in four domains.

South and Southeast Asian countries are identified based on the South Asian Association for Regional Cooperation (SAARC) and the Association of Southeast Asian Nations (ASEAN). The first author developed the search terms, selected databases, and conducted the initial search and screening process in consultation with a senior health librarian agreed upon by all authors. Keywords were searched connecting Boolean operators OR/AND. The searches were restricted to title, abstract, and subject due to a large number of studies obtained by the initial search. The search terms were modified for each database as necessary (Appendix A). In addition, a search in Google Scholar and the reference lists of the included studies were conducted for any additional references that could have been overlooked/missed during the database searches.

### 2.4. Data Extraction

The first reviewer (S.R.F.) extracted data using a standardised data extraction form and the second reviewer (L.E.) confirmed the extracted information. The data extraction form was developed using Joanna Briggs Institute manual [23] and grouped into three sections. The first section was related to characteristics of the studies such as the title, citation, authors, year of publication, and source of publication. The second section documented information on the methodology, study objectives, and demographics of the study participants. In particular, information related to the objectives, study design, study setting, type of participants, age, sex, type of data, data collection instrument, sampling procedure, sample size, and data analysis procedure were recorded in this section. The third section documented the main findings, limitations of the study, relevant additional information, and recommendations.

### 2.5. Quality Assessment

The quality of the 16 articles were assessed using the criteria of the Mixed Method Appraisal Tool (MMAT) [24], which is designed for the methodological quality appraisal of systematic mixed-method reviews and quantitative, qualitative and mixed-method studies. Two screening questions including the inclusion of a clear research question and data addressing the research question were used for all types of studies. The quantitative articles were assessed using the descriptive and non-randomised categories of the MMAT. The quality of the qualitative and mixed-method study was assessed using the qualitative and mixed-method categories of the MMAT, respectively. Each criterion carried one point in the assessment of a study, with a maximum score of 7 along with the points of 2 screening questions. As per this assessment tool, articles scoring 1–3 points were considered as “low quality”, 4–5 as “medium quality”, and 6–7 as “high quality”. The quality assessment of the included studies was independently carried out by the lead author (S.R.F.) and another team member (M.S.I.).

### 2.6. Data Synthesis

Results of the review were synthesised using an integrated mixed-method synthesis approach to summarise all quantitative, qualitative, and mixed-method data into a single combined synthesis [25]. All data were analysed and integrated, and the results across all studies presented as a narrative summary. Although the majority of reviewed studies used a cross-sectional design, due to substantial differences among individual studies in terms of settings, samples, populations, context, health assessment tools, and outcome assessment statistical combination of results (meta-analysis) are not possible [26]. Therefore, all data were analysed and integrated, and the results across all studies are presented as a narrative summary.

## 3. Systematic Review Results

A total of 796 studies were found through electronic databases (n = 779) and hand searching (n = 17), of which 16 fulfilled the inclusion criteria after a full-text evaluation. All the articles were imported into EndNote library version X9. Two independent reviewers (S.R.F. and K.U.) involved in the study selection procedure, and disagreement was resolved in consultation with other team members. Figure 1 illustrates the PRISMA flow diagram with the evaluation and selection procedure of the chosen studies.

### 3.1. Study Characteristics

The study characteristics are outlined in Table 2. Of the studies included in this review, 14 (88%) were quantitative, 1 was qualitative, and 1 study used a mixed-methods design. The most common research method was the quantitative approach, which included 10 quantitative descriptive studies and four quantitative non-randomised studies. All the studies were conducted in lower-middle-income countries: Nepal (n = 6) [27,28,29,30,31,32], Pakistan (n = 4) [33,34,35,36], India (n = 2) [37,38], Indonesia (n = 2) [39,40], Sri Lanka (n = 1) [41], and Bangladesh (n = 1) [42]. Among these, Nepal and Bangladesh have recently risen up from low-income country status to lower-middle-income countries. The studies assessed exposure to disasters caused by three different types of natural disasters, most frequently, exposure to earthquakes (75%), tsunamis (19%), and cyclones (6%). Women represented almost two-thirds of the study populations presented in the studies. The majority of the studies (14/16) included disaster-affected men and women as the study participants and the other two reported women in particular. One study did not report the age range, although the study included adult community members [31]. Most of the studies (n = 8) were published between 2018 and 2019, one in 2020, and seven between 2010 and 2012.

### 3.2. Quality Assessment

The results of the quality appraisal are presented in Appendix A. The quality appraisal procedure resulted in 14 of 16 articles being rated as high quality and the other two medium quality. No studies were excluded based on the quality assessment scores. The mixed-method study was assessed as medium quality article as per the MMAT. This outcome was due to its failure to report the quantitative and qualitative methods of data collection distinctly, inconsistency between results, and the lack of a rationale for the need for a mixed-method study.

### 3.3. Mixed Method Synthesis

Based on our individual study aim’s, an evidence table (see Table 3) was developed aggregating findings to synthesise aspects focused on the women’s health impact and underlying risk factors following natural disasters. 

The quantitative, qualitative, and mixed method data were extracted and viewed to identify common themes among the retained studies. Three core themes were identified and are presented in Table 4.

### 3.4. Health Impact of Natural Disasters on Women

Three sub-themes were identified within the literature considering the variation of health outcomes. These include (i) physical health outcomes, (ii) mental health outcomes, and (iii) post-traumatic stress disorder. As mentioned, Table 3 illustrates the main risk factors affecting women’s health that have been cited in this review.

#### 3.4.1. Physical Health Outcomes

The findings from the review provide evidence that natural disasters have a direct impact on the physical health of women. Two studies conferred that women experienced physical injury and disability following earthquakes [27,39]. Among the included studies, a cross-sectional study conducted in Nepal, the most earthquake-affected area in South Asia, identified physical injury and disability as an outcome of earthquakes. A high percentage (80%) of the survivors had upper extremity injuries followed by lower extremity injuries (31%), head injuries, and miscellaneous injuries such as to the rib or clavicle. Multiple injuries and spine or spinal cord injuries were also seen in a significant percentage (62%) of participants. In the study by Bimali et al., (2018), the highest percentage (85%) of disability was observed in women even though there were more men included in the study [26]. Similarly, a prospective cohort study reported consistent and significant disability scores among injured women survivors in Indonesia [39]. Women with the highest disability scores reported pain and discomfort as negatively impacting their quality of life [39].

#### 3.4.2. Mental Health Outcomes

The majority (n = 8) of retrieved studies strongly identified the association between depression, anxiety, and common mental disorder (CMD) in women following natural disasters [28,31,32,33,34,35,36,37,42]. All these studies measured/observed significant scores of depressions (mean range of score 9.88–29.14), anxiety (10.97), and CMD (General Health Question score of 4 and above, mean score of distress 11.2) in disaster-affected participants using a variety of scales including the Depression Anxiety Stress Scale (DASS)-21, the Patient Health Questionnaire-9 (PHQ-9), the Beck Depression Inventory-II (BDI-II), the General Health Questionnaire (GHQ 12), and the Centre for Epidemiological Studies Depression Scale (CES-D). More than half of the reviewed studies (n = 10) associated with male and female participants revealed that women were more vulnerable in regard to negative psychological or mental health outcomes compared to men [27,29,30,31,32,34,35,36,37,38]. For example, one study found that women had almost twice the rate of CMD than men, while women over 30 years of age had approximately two times greater odds of experiencing CMD than those younger than 30 years [37]. Most (n = 5) of these studies identified depression as a common illness in disaster-affected women followed by anxiety, CMD, and other psychosocial and mental health problems. In addition, higher mean of depressive symptoms (10.8 and 29.14, respectively) in women after natural disasters were confirmed by two specific studies on women [41,42]. The qualitative study observed psychosocial and mental health problems such as forgetfulness, tiredness, loss of concentration, restlessness, and isolation in older people, largely women, following an earthquake [28].

#### 3.4.3. Post-Traumatic Stress Disorder (PTSD)

In addition to the reported mental disorders of anxiety, depression, and CMD, almost half of the studies reported that disaster-affected women are exposed to PTSD [29,30,32,35,36,38,40]. In post-disaster circumstances, PTSD is the highest recorded health issue compared to other mental health problems (i.e., depression, anxiety, CMD) for women documented through retrieved studies [29,30,32,35,36,38,40]. Studies used different recognised measures to determine the level of PTSD such as Clinician Administered PTSD Scale version 5, the standard PTSD symptoms checklist, the PTSD checklist civilian version (PCL-C), the Impact of Event Scale-Revised (IES-R), and the Traumatic Stress Symptoms checklist. Two studies showed intrusion symptoms for example (3.24 ± 0.71) were high among the 91% survivors with PTSD [30,40]. In addition, all these cross-sectional studies showed that females experienced a greater level of mental health complications related to PTSD compared to males after earthquakes and tsunamis in Nepal, Pakistan, and India [29,30,32,35,36,38,40].

### 3.5. Risk Factors Affecting Women’s Health Following Natural Disasters

Our review identifies several types of risk factors for women’s negative health outcomes in post-disaster circumstances. Four sub-themes are identified: (i) socio-demographic risk factors, (ii) disaster exposure, (iii) post-disaster factors, and (iv) pre-existing risk factors.

#### 3.5.1. Socio-Demographic Risk Factors

All retrieved studies have reported a significant association between socio-demographic factors and women’s health following natural disasters. As Table 3 illustrates, twelve socio-demographic factors were identified throughout all studies. The reviewed literature confirmed that being female was the foremost risk factor following disasters. The majority (63%) of the studies reported that being either adult and/or being an older woman was linked to the highest risk for PTSD, depression, anxiety, CMD, disability, and poor physical, and psychosocial and mental health problems [27,28,29,30,31,36,37,38,40,41]. Alternatively, three studies showed the lower age group (18–30) of women were the significant risk factors for depression and physical injury [33,39,42]. Similarly, a majority (57%) of studies found women with no formal education or lower level of education were the highest risk of developing PTSD, depression, CMD, disability, and poor physical health consistently [29,30,32,34,36,37,39,40,41]. Other significant risk factors identified in the literature were poverty or low income, followed by marital status (i.e., being single or widowed), religious minorities, having no children, ethnic groups, and socially disadvantaged groups [28,29,31,32,36,37,38,41,42]. Studies conducted in India and Bangladesh reported women as the only income earner of the family and rural residence is a significant risk factor for depression and PTSD, respectively, post-disaster [38,42].

#### 3.5.2. Disaster Exposure

Exposure to a disaster was the most notable risk factor for disaster-related health problems for any disaster survivor. Among the reported disaster variables throughout the literature, the most common (38%) recorded reasons for disaster-affected women’s health adversities were physical injury and resource loss including financial, food, shelter, and property [28,29,30,32,35,37,38,39,42]. Studies showed that these two disaster exposures are significantly associated with disability, depression, CMD, psychosocial and mental health problems, and PTSD [28,29,30,32,35,37,38,39,42]. Studies (n = 5) from India and Pakistan showed humanitarian loss (i.e., death of family members, close relatives, or friends) and displacement (i.e., living in tent) as the second highest causes for women’s PTSD and CMD after a disaster [29,34,35,36,37,38]. Two studies from Pakistan found women who experienced disability and PTSD more likely remained at home during the earthquake [34,36]. The women in India [38] and Nepal [30] scored high levels of PTSD as they felt their lives were threatened and experienced panic during a disaster. Other causes of PTSD for these women were related to difficulty in communication, separation from family or when relatives were injured, and if they witnessed the death of family or community members [29,34,36,38]. In addition, and within the context of earthquakes, greater levels of anxiety and depression were recorded in Nepalese women who went to traditional healers as their source of health information [31], while Indonesian women recorded significant PTSD scores if they could not access public health centres [40].

#### 3.5.3. Post-Disaster Risk Factors

Post-disaster risk factors were mostly associated with PTSD, anxiety, depression, and CMD, and psychosocial and mental health problems were evident in 63% of the reviewed studies. The most common (13%) post-disaster risk factors were loss of job or income generation activities, low social support, and negative religious coping [29,36,37,38]. A study from Nepal showed that inadequate access to health care facilities due to poverty made women more vulnerable to anxiety and depression [31]. Women recorded significant depressive symptoms when they experienced poor physical health and became dependent on others, and although they survived, they lived with the fear of re-experiencing a disaster [32,33,41]. In addition, a study from Bangladesh revealed that depressive symptoms were dominant when women were the income-earner yet were unable to work due to physical injury or poor physical health [42]. Indian women recorded high PTSD scores associated with inadequate resources after a disaster, lack of mental health support systems, involvement in rescue work, and substance abuse [38]. Moreover, negative religious practices such as a feeling of being punished by God for one’s sins or lack of spirituality was significantly associated with higher symptoms of depression and PTSD in Nepal and India [36,37]. The qualitative study also found that older women were blamed for the negative events in society, and this accusing practice severely affected their psychosocial and mental health [28].

#### 3.5.4. Pre-Existing Risk Factors

Nearly half of the reviewed studies (n = 6) reported how pre-existing disaster factors of low and lower-middle-income countries impact women’s health following natural disasters [28,29,37,38,40,41]. Studies showed Sri Lankan and Nepalese women more exposed to family adversities, stress and violence before disasters were generally vulnerable to PTSD, and poor physical, psychosocial, and mental health problems in post-disaster [28,29,41]. In addition, some studies (n = 4) identified those suffering from chronic illness and with a family history of mental illness were more vulnerable to developing PTSD and CMD [29,37,38,40]. In addition, CMD and PTSD in women was associated with low quality of marital relationships and history of substance use [37,38].

### 3.6. Association between Physical and Mental Health

Physical and mental wellbeing are interrelated—poor physical health affects mental health and vice versa. Studies (n = 2) presented direct correlation of physical and mental health, and some of the studies identified poor physical health (i.e., ill health, physical injury, and disability) as a significant risk factor for mental health problems after a disaster [39,41]. The prospective cohort study in Indonesia showed that injured earthquake survivors reported significantly higher proportions of severe complaints of co-morbidities (i.e., acute symptoms and chronic diseases) as compared to the non-injured survivors group. Among the injured group, the lowest average quality of life attribute (QLA) score was attributed to pain, depression, and anxiety throughout the study period [39]. However, the longitudinal Sri Lankan study of women described the existence of physical and mental health problems in an inverse relationship. After tsunamis, the average depressive symptoms showed a decline, whereas poorer physical health increased over the time of the study [41].

## 4. Discussion

The review identified a variety of physical and psychological health consequences after natural disasters and risk factors affecting women in South and Southeast Asia based on 16 empirical peer-reviewed research articles. The review included studies conducted among twenty SAARC and ASEAN countries. Fourteen were from South Asia, covering Nepal, Pakistan, India, Sri Lanka, and Bangladesh, and the remaining two were from Indonesia and the Southeast region. The studies showed that there are variations in disasters between countries, which is influenced by geography [3]. The review identified that Nepal-, Pakistan-, and Indonesia-based studies were primarily on earthquakes, while India- and Sri Lanka-based studies were on tsunamis, and cyclones were the focus of studies in Bangladesh. Although there is a difference in the type of disasters from country to country, the health impacts reported post-disasters are similar.

Our synthesis of the relevant published studies provided strong evidence for significant levels of health problems within disaster-affected women, which is consistent with previous evidence [11,43]. In support of the findings of other studies, the review showed that disability from injury can significantly impact women’s physical health. For example, a high prevalence of pelvic fractures and inflammation among women were reported after the Wenchuan earthquake in China [44,45]. Due to these injuries, survivors lost their mobility and became dependent on others, which led them to experience a poorer quality of life [7,11,46]. These physical health vulnerabilities impact women’s mental health and in the long term and can lead to the development of PTSD. The majority of reviewed studies (n = 13) also showed that women faced different types of mental health problems including PTSD, depression, CMD, and anxiety after disasters in accordance with the findings of prior studies in Iran, Japan, and the USA [11,47,48], and PTSD of women after the Haiti and Wenchuan earthquakes [17,18].

Disaster-affected women are often disadvantaged and vulnerable because of biological, physical, socio-demographic, cultural, and economic factors [11]. In the context of disasters, a total of 35 risk factors for negative health outcomes in women were explored and categorised under four groups (i.e., socio-demographic, disaster exposure, post-disaster, and pre-disaster/pre-existing). Six risk factors were dominant in six to ten studies and seven risk factors were explored in three or more studies. Similar to other studies, the review showed that socio-demographic factors such as adult/older age, having no formal education, severe economic status, divorced/widowed, and low social support were independent risk factors for health problems in most women [11,17,47]. Women’s socio-demographic profile influences their economic resources, social status, social networks, and health behaviour. Therefore, not all women are vulnerable after a disaster, but women with a low socio-demographic profile reported health problems worldwide [10,11,17,47,48]

The most reported disaster exposures affecting women’s mental health were related to physical injury, resource loss (e.g., loss of food, property, damage house), and death of close family members. Higher post-traumatic symptoms were generally found to be worse for those with physical injury and the loss family members in the aftermath of disasters [49,50]. It is understandable that with loss of property, individuals face persistent financial problems and difficulties with living arrangements. In addition to these, loss of employment or income-generating activities constituted ongoing stress that could adversely affect mental health. In most cases, the affected women were too poor to pursue treatment and did not treat their diseases seriously [7,11]. Findings also indicated that being an income earner was a risk factor for post-disaster health problems, which was the opposite to other studies showing that economic inactivity was associated with increased rates of depression [51,52]. However, women found that their mental health was improved when their livelihood needs were met.

The majority of studies reported that post-disaster risk factors were relevant to coping strategies and disaster management systems of countries. Low social support is significantly associated with health problems after disasters whereas strong support from family and the community and access to health care facilities appear to safeguard people from the negative impact of traumatic events. The right support is also important for preventing mental disorders and helping individuals adapt to disasters. For instance, women affected by Hurricane Katrina benefitted from social support four years after the storm [53]. Some studies revealed, however, that women with poor socio-demographic backgrounds appeared to have a lower capacity of managing and coping with both pre-disaster and post-disaster situations [11,43]. Additionally, pre-existing family adversities, stress, chronic illness, and low-quality marital status potentially make women vulnerable to health problems.

### 4.1. Policy Implication

The review emphasises the implications of the effective emergency response and recovery in the post-disaster situation along with disaster preparedness and mitigation plans in the pre-disaster context. Both are essential for helping and improving the physical health, mental health, and wellbeing of women [54,55]. The most important issue is to mitigate the impact of disasters; disaster management should be preventive rather than responsive [55]. Therefore, pre-impact conditions, hazard mitigation, and preparedness efforts needed to be prioritised in policy and practice [54,56]. In addition, policies needed to prioritise effective interventions focusing on livelihood support to women to promote health and resilience after a disaster [55]. Social policies including building houses, roads, and river barriers are essential for disaster-prone residents of rural and remote communities. Older disaster victims are more at risk of negative health outcomes, evidenced by low social support, economic dependency, re-experiencing disasters, pre-existing chronic health conditions, and feelings of distress during a disaster [30,33,34,39,57]. Emotional support is identified as an important factor for promoting the health of elderly disaster victims [58]. Therefore, elders’ health necessities must be considered with particular attention in disaster management policy.

### 4.2. Clinical Implications

This review highlights that overcoming the mental illness experienced by disaster-affected women requires urgent clinical attention and implication in the South and Southeast Asia region. The WHO (2007) recommended disaster mental health guidelines for disaster-prone countries, which are already implemented by developed countries such as the United States, Australia, and Japan [14,15]. In any health emergency situation, the role of emergency medical teams including doctors, nurses, and community health workers are essential to reduce deaths, disabilities, and diseases [59]. Clinicians and mental health professionals in communities receiving disaster survivors could facilitate individual disaster recovery through understanding their individual stressors and circumstances, which was found in this review. Disaster-affected individuals will benefit from the collaboration between disaster management and the healthcare sector. In disaster-prone countries, health professionals’ active involvement in pre-disaster and during disaster health risk assessment and post-disaster recovery programmes can mitigate the health impacts of vulnerable populations, such as women. It is recommended that the availability of access to female health professionals can reduce mental illness. For example, following the Nepal earthquake, women who went to female community health workers reported lower anxiety and depression symptoms compared to those who went to other services [31]. 

## 5. Conclusions

Increasing natural disasters is a worldwide issue, and is associated with negative physical and mental health issues, particularly among women. The results of this systematic review suggest that the adverse impacts of natural disasters extend beyond the explicit consequences of other damages in terms of women’s health. The findings of this review support the view that women’s health problems are a result of a complex interplay between pre-disaster, disaster, and post-disaster factors. The most-cited risk affecting disaster survivors’ health included being female, being from a poor socio-economic background, resource loss, humanitarian loss, negative coping, dependency, and loss of income. In the context of low- and lower-middle-income countries, improved communication plans need to be developed for disaster-prone residents, who need free access to health information/facilities. Future health policy and disaster management consultants need to consider how best to address various health problems and risk factors in the most effective way. In addition, policy should help unprivileged women of developing countries to recognise that a mental health problem exists, and to believe that seeking help may be beneficial in promoting their health and wellbeing. Facilitating women’s capacity of managing and coping with both pre-disaster and post-disaster situations is crucial. Moreover, the identification of health impact and risk factors affecting women’s health is essential for reducing the burden of health problems. Therefore, this research is an important step in synthesising some of these important factors and outlining possible suggestions for prevention, as well as affording guidance for future research.

### 5.1. Limitations

As with all reviews, there are inherent limitations in this study; hence, it is important to acknowledge the limitations of this review. First, reports, grey literature, and books, which may be based on primary data, were not included. Therefore, there is a possibility that these sources may have yielded additional results. Second, searches in additional databases without search parameters could gather more information which could strengthen the generalisability of the findings of this review. Third, some relevant studies may have been missed due to strictly following the inclusion criteria that focused on adult women. For example, no study was retrieved from the Philippines in this review, which is a significant disaster-affected country of Southeast Asia. Fourth, although reproductive health was included as a search term, no studies on women’s reproductive health challenges after disasters were identified in this review that met the inclusion criteria. The majority of reviewed studies used a cross-sectional design, which may prevent researchers from making strong inferences about the causation of women’s health impacts and risk factors reported in the studies, as the data observe the study population at only one point in time. The review did, however, identify women’s health impacts following natural disasters, the prevalent risk factors, and the current knowledge gaps.

### 5.2. Recommendation for Future Research

A variety of the health assessment tools were used in the reviewed studies. A greater consistency in assessment tools used to examine the extent of physical and mental health difficulties may be beneficial for future research. We also recognise that further work is needed in this area, particularly in relation to research that yields high-quality evidence, such as studies that utilise non-probability sampling to enhance the quality of the work and the confidence in the evidence provided. Further qualitative and mixed method research needs to be carried out. More research related to the aim of this review was conducted in Nepal and Pakistan than the other countries; thus, more is generally known about the health impact and risk factors of these survivors. Notably, although Bangladesh is the fifth most disaster-affected country in the world [60] and records floods, cyclones, and riverbank erosions every year [12], only one study retrieved from Bangladesh identified women’s mental health outcomes after disasters. Given the large number of disasters, there is a scarcity of disaster research focused on the health impact of disasters on women in Bangladesh.

## Figures and Tables

**Figure 1 ijerph-18-11068-f001:**
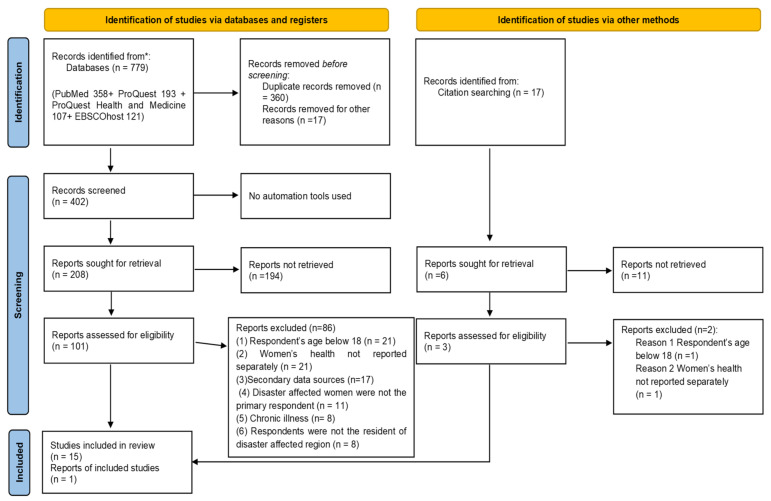
PRISMA flow diagram of included studies.* According to PRISMA * means: consider reporting the number of records identified from each database rather than the total number across all databases.

**Table 1 ijerph-18-11068-t001:** Search terms.

Population	Women OR Woman OR Female OR Females
Interest	health (health (vulnerability OR vulnerabilities OR risk OR hazard OR hazards OR hazardous OR psychological OR mental OR physical OR emotion OR emotional OR psychosocial OR reproductive OR sexual))
Context_1_	natural disasters (“natural disasters” OR “natural disasters” OR “natural calamities” OR “natural calamity” OR flood OR floods OR flooding OR volcano OR volcanoes OR volcanic OR earthquake OR earthquakes OR cyclone OR cyclones OR hurricane OR hurricanes OR drought OR droughts OR tornado OR tornadoes OR landslide OR landslides OR mudslide OR mudslides OR (“wild fire” OR “wild fires” OR “wildfire” OR “wildfires” OR bushfire OR bushfires)
Context_2_	“south asia” OR “southeast asia” OR bangladesh OR srilanka OR sri Lanka OR india OR bhutan OR nepal OR pakistan OR maldives OR afghanistan OR brunei OR burma OR mayanmar OR cambodia OR timo-leste OR indonesia OR laos OR malaysia OR phillippines OR singapore OR thailand OR vietnam

**Table 2 ijerph-18-11068-t002:** Characteristics of selected studies.

Reference/Location	Sample Size/Age/Gender	Context/Type of Participants	Study Design/Methods	Quality
Bimali et al., (2018), Nepal [27]	Total sample: 199 Female: 25 to 80 Male: 9 to 79 Male: 65% Female: 35%	Earthquake Individuals with physical disabilities caused by the earthquake	Cross-sectional descriptive study Quantitative	HQ
Adhikar et al., (2018), Nepal [28]	Total sample: 30 Age range not reported Women: 7 Men: 23	Earthquake Disaster-affected older people and key informant	Qualitative	HQ
Dahal et al., (2018), Nepal [29]	Total sample: 535 Age: 18+ Female: (n = 247, 46.2%), Male: (n = 288, 53.8%)	Earthquake Earthquake experienced survivors	Cross-sectional study Quantitative	HQ
Baral et al., (2019), Nepal [30]	Total sample: 291 Age: 20 and above Female: (n = 125, 43%) Male: (n = 166, 57%)	Earthquake Earthquake experienced adult survivors	Cross-sectional descriptive study Quantitative	HQ
Powell et al., (2019), Nepal [31]	Total sample: 750 Age range not mentioned Female: (n = 532, 70.9%) Male: (n = 218, 29.1%)	Earthquake Disaster-affected individuals	Cross-sectional study Quantitative	HQ
Schwind et al., (2019), Nepal [32]	Total sample: 238 Age: 18–85 Female: (n = 145, 65%) Male: (n = 78, 35%)	Earthquake earthquake experienced adult survivors	Cross-sectional study Quantitative	HQ
Suhail et al., (2009), Pakistan [33]	Total sample: 125, response rate of 98.45%. Age: 18–70 Women: 73 Men: 52	Earthquake Earthquake survivors	Mixed-method	M
Ahmad et al., (2010), Pakistan [34]	Sample size: 1st wave: 44, 2nd wave: 51 Mean age: 1st wave; 35.3, 2nd wave: 31.2 1st wave Female 12 (27.3%), Male 32 (72.7%) 2nd wave Female: 0 Male: 51 (100.0%)	Earthquake Earthquake survivors	Cross-sectional study Quantitative	M
Naeem et al., (2011), Pakistan [35]	Total sample: 1298 Age: 18+ Female: (n = 779, 60.3%) Male: (n = 512, 39.7%)	Earthquake Disaster-affected residents in earthquake area	Cross-sectional study Quantitative	HQ
Feder et al., (2012), Pakistan [36]	Total sample: 200 Age range not reported but study on adult survivors Female: 39 Male: 161	Earthquake Adult earthquake survivors	Cross-sectional study Quantitative	HQ
George et al., (2012), India [37]	Total sample: 533 Age: 18+ Female: (n = 303, 57%) Male: (n = 229, 43%)	Tsunami Tsunami-affected residents	Cross-sectional study Quantitative	HQ
Pyari, et al., (2012), India [38]	Total sample: 485 Age: 19 to 81 Female: (n = 178) Male: (n = 121)	Tsunami Tsunami survivors with PTSD	Quantitative	HQ
Sudaryo et al., (2012), Indonesia [39]	Injured: 184 Non-injured: 93 Age: 18+ Injured Men: 53, Women: 131 Non-injured Men: 22, Women: 71	Earthquake Adult injured earthquake survivors	Cohort study Quantitative	HQ
Aurizki et al., (2020), Indonesia [40]	Total sample: 152, response rate 100% Age: 60 and above Females (n = 113, 74%) Males (n = 39, 26%)	Earthquake Adults experienced or witnessed the disaster directly	Cross-sectional study Quantitative	HQ
Wickrama et al., (2011), Sri Lanka [41]	Sample size: wave 1: 195, wave 2: 160 Age: 29 to 60, Female	Tsunami Tsunami-exposed mothers	Longitudinal Study Quantitative	HQ
Mamun et al., (2019), Bangladesh [42]	Total sample: 111 Age: 18 and above Female	Cyclone Women, permanent residents of the Cyclone Mora-affected village	Quantitative	HQ

**Table 3 ijerph-18-11068-t003:** Risk factors and health outcomes of women of selected studies.

Risk Factors	Health Outcomes of Women Following Disasters	Total Studies (and %) Naming These Stress Affecting Women’s Health
Socio-demographic risk factors		
Adult Age [27,28,29,30,31,36,37,38,40,41]	PTSD, depression, anxiety, CMD, disability, poor physical health, depressive symptoms, psychosocial and mental health problems	10 (63%)
No education/lower level of education [29,30,32,34,36,37,39,40,41]	PTSD, poor physical health, depressive symptoms, depression, disability from injury, CMD	9 (57%)
Poverty/low income/economic hardship [28,31,37,39,40,41]	Anxiety, poor physical health, depressive symptoms, psychosocial and mental health problems, disability from injury, PTSD, CMD	6 (38%)
Marital status (single/divorced/widowed) [32,36,37,38]	PTSD, depression, CMD	4 (25%)
Lower age group (18–30) [33,39,42]; religious minorities [29,32,37];	Depression, injured, PTSD, CMD	3 (19%)
Having children or no children [38,42]	Depression, PTSD	2 (13%)
Family structure, housing type [37], socially disadvantaged group [32]; being an income earner [42]; rural residence [38]	Depression, CMD, PTSD	1 (7%)
Disaster exposure		
Disaster related physical injury [28,30,32,38,39,42]	Depression, PTSD, psychosocial and mental health problems	6 (38%)
Resource loss (financial loss or loss of food, shelter, property)/completely damaged house [29,32,35,37,38,39]	PTSD, CMD, depression, disability	6 (38%)
Humanitarian loss [29,34,36,37,38]	PTSD, CMD	5 (32%)
Relocation or displacement/living in tent [34,35,38]	PTSD, CMD	3 (19%)
Distance from epicentre [34,36]; position (staying home) during earthquake [27,35]	PTSD, CMD, disability from injury	2 (13%)
Source of information for health issues [31]; public health centre utilisation [40]; injury to family members [38]; difficulty in communication, witnessing death [29]; threat to life/panic during disaster; community destruction; separation from family [38]	PTSD, anxiety, depression	1 (7%)
Post-disaster risk factors		
Loss of job/income generation activities [29,37]; low social support [29,38]; negative religious coping [36,37]	PTSD, CMD	2 (13%)
Poor physical health [41]; access to healthcare facility [31]; dependence on others [32]; absence from work/work absenteeism [42]	Depression, anxiety, poor physical health, depressive symptoms	1 (7%)
Fear of re-experiencing disaster [33]; blaming [28]; inadequate resources, mental health support system, involvement in rescue work; substance abuse [38]	Depression, psychosocial and mental health problems, PTSD	1 (7%)
Pre-disaster risk factors		
Pre-disaster stress and exposure to violence, family adversities/conflict (secondary stressors) [28,29,41]; chronic illness [37,38,40]	PTSD, CMD, poor physical health, depressive symptoms, psychosocial and mental health problems	3 (19%)
Exposure of family history of mental illness [29,38]	PTSD	2 (13%)
Low quality marital relationship [37]; poor socioeconomic background; history of substance use [38]	PTSD, CMD	1 (7%)

**Table 4 ijerph-18-11068-t004:** Schematic of themes.

Themes		Sub-Themes
3.4	Health impact of natural disasters on women	3.4.1 Physical health outcomes 3.4.2 Mental health outcomes 3.4.3 Post-traumatic stress disorder
3.5	Risk factors affecting women’s health following natural disasters	3.5.1 Socio-demographic risk factors 3.5.2 Disaster exposure 3.5.3 Post-disaster factors 3.5.4 Pre-existing risk factors
3.6	Association between physical and mental health	-

## Data Availability

Data sharing is not applicable to this article.

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
