# Peer review of "Health Impact and Risk Factors Affecting South and Southeast Asian Women Following Natural Disasters: A Systematic Review"

_ijerph, 2021, doi:10.3390/ijerph182111068_

Round 1
Reviewer 1 Report
The research topic seems interesting and novel.
A clear and concise introduction is presented.
The methodology is rigorous, well developed and concrete.
The sections studied are developed in a consensual manner.
In the Discussion section (line 416-434), it seems to be the results section again. It is repetitive, providing no discussion or new references.
The same line 426-429 seems to deal with part of the conclusion,
a situation that occurs again between lines 432-434.
From line 453 to 466, it is recommended to be presented as a separate section from the discussion. After the conclusions and before the recommendations.
Author Response
Dear Reviewer,
The authors are very grateful to the reviewer and to the editor for their suggestions to improve the quality of the article. Please see the point-by-point responses to the comments below. All authors contributed to the revision of the manuscript.
Kind regards,
Syadani Riyad Fatema, PhD student, BSS (Hons.), MSS.
Associate Professor Leah East, RN, BN(Hons), PhD, GradCertAP
Dr. Md Shahidul Islam, PhD, MSC, MSS, BSS (Hons.), DTMH, AFACHSM
Professor Kim Usher, AM, RN, PhD, FACMHN, FACN

Reviewer 2 Report
Dear authors.
this is an interesting piece of work with the aim to identify and critique the evidence of the impact of natural disasters on the physical and mental health of women, and the underlying causes/risk factors affecting their health in the countries located in South and Southeast Asian regions. Through identification of the impact and risk factors, this study can assist and improve preparedness for disasters and the implementation of appropriate plans to potentially reduce the impact of disasters on women’s health and wellbeing.
I was wondering why this review was limited between 2008 to 2021. Is there a reason for that? I would suggest you extend the research before 2008.
The majority of reviewed studies used cross-sectional design which may have prevented researchers for making strong inferences about causation of women’s health impacts and risk factors reported in the studies, as the data observe the study population at only one point in time. This may have limited the importance of the results.
Could a meta-analysis be conducted on the basis of this systematic review.
Author Response

(The authors gave the same response as above.)

Reviewer 3 Report
The authors note that the results of the very thorough and appropriate search process were very heterogenous, which makes intepretation/synthesis of the studies quite difficult to achieve. The studies individually look at impacted individuals by and large, and so particularly when it comes to qualitative studies, have a bias towards finding those who suffered from a particular natural disaster. There is no sense, then, collectively of those who did NOT suffer from the disaster. So the chief weakness of what is otherwise a very well conducted systematic literature review is twofold: the banality of the findings (e.g. that socioeconomic disadvantage lead to greater impact of disasters on women, and that pre-existing adversity leads to subsequent adversity) and a superficiality in the process of synthesis by the authors. Just to focus on this latter criticism, which is the more serious one, in a sense the authors have conducted what is a systematic quantitative literature review, and they focus on "almost half" of studies have found x, and a majority have found y. Yes, there is a degree of 'garbage in, garbage out' to the world of systematic review, but I suspect if instead of focusing on these four databases, and these search parameters, the authors had conducted a more exhaustive approach to finding studies that did population analyses post-disaster, or only selected studies that use structured or random sampling to ensure that the results were generalisable to the population, then the study would have been ultimately more impactful.
Let's look at one of these studies, in many senses one of the better studies included, by George et al. The study is large but cross-sectional, and non-probability sampling was used--and no further details are given. Individually, in the wake of a disaster, non-probabilistically sampling survivors may not tell us a great deal about the survivors health, and conducting statistical analysis on respondents may not improve our insight as to factors related to mental health (as these authors do). The majority of the studies are less capable of arriving at results that can be generalised. The Indonesia prospective cohort study is one outlier, but my general point is that it is difficult to see the 16 studies as providing "strong evidence".
To some degree I speak from 'experience' in this regard, having spent an extended time over many years working in one of the world's great disaster zones, interviewing survivors and observing impacts. When I asked survivors about the gender issue, repeatedly it was clear that the majority of women and children had died in the 'front-line' of this disaster, and it was clear also why they had died. If one were to have done a random sampling of survivors in this region, post disaster, the resultant sample would have clearly contained very few women. This is an extreme case, an extraordinary catastrophe, but I think it showed the greater vulnerability of women in south Asia just in the way that this review is showing... I note that Sudaryo who conducted a study with both injured and non-injured earthquake survivors showed similar a much greater injury rate amongst women. So papers that do not have strong sampling approaches cannot be regarded as strong evidence, regardless of whether they are qualitative or quantitative in approach.
Author Response

(The authors gave the same response as above.)

Round 2
Reviewer 2 Report
this is an interesting piece of work with the aim to identify and critique the evidence of the impact of
natural disasters on the physical and mental health of women, and the underlying causes/risk factors
affecting their health in the countries located in South and Southeast Asian regions. Through
identification of the impact and risk factors, this study can assist and improve preparedness for
disasters and the implementation of appropriate plans to potentially reduce the impact of disasters on
women’s health and wellbeing.